health and disease and epidemiology

COVID-19, travel restrictions, Newfoundland and Labrador, importations, epidemic model, branching process

**Author for correspondence:**
Amy Hurford
e-mail: ahurford@mun.ca

# Modelling the impact of travel restrictions on COVID-19 cases in Newfoundland and Labrador

Amy Hurford[1,2], Proton Rahman[3] and
J. Concepción Loredo-Osti[2]

[1]Department of Biology, and [2]Department of Mathematics and Statistics, Memorial University, St John's, Newfoundland and Labrador, Canada A1B 3X9
[3]Faculty of Medicine, Memorial University, St John's, Newfoundland and Labrador, Canada A1C 5B8

AH, 0000-0001-6461-5686

In many jurisdictions, public health authorities have implemented travel restrictions to reduce coronavirus disease 2019 (COVID-19) spread. Policies that restrict travel within countries have been implemented, but the impact of these restrictions is not well known. On 4 May 2020, Newfoundland and Labrador (NL) implemented travel restrictions such that non-residents required exemptions to enter the province. We fit a stochastic epidemic model to data describing the number of active COVID-19 cases in NL from 14 March to 26 June. We predicted possible outbreaks over nine weeks, with and without the travel restrictions, and for contact rates 40–70% of pre-pandemic levels. Our results suggest that the travel restrictions reduced the mean number of clinical COVID-19 cases in NL by 92%. Furthermore, without the travel restrictions there is a substantial risk of very large outbreaks. Using epidemic modelling, we show how the NL COVID-19 outbreak could have unfolded had the travel restrictions not been implemented. Both physical distancing and travel restrictions affect the local dynamics of the epidemic. Our modelling shows that the travel restrictions are a plausible reason for the few reported COVID-19 cases in NL after 4 May.

## 1. Background

In response to the COVID-19 pandemic, travel restrictions have frequently been implemented [1], yet the efficacy of these restrictions has not been established. Some previous studies consider the impact of international travel restrictions [2–5], but there is a paucity of studies considering restricted travel within a

nation [4] making the implementation of travel restrictions controversial for public health authorities [1]. Furthermore, the impact of travel restrictions on reducing COVID-19 spread is interwoven with the impacts of other public health measures. For example, the spread of imported cases depends on compliance with self-isolation directives for travellers, local physical distancing and mask wearing. Travel restrictions were implemented in Newfoundland and Labrador (NL) on 4 May 2020, such that only NL residents and exempted individuals were permitted to enter the province. We use a mathematical model to consider a 'what-if' scenario: specifically, 'what if there were no travel restrictions?', and in doing so, we quantify the impact that the travel restrictions had on the number of subsequent COVID-19 cases in NL.

Mathematical models appropriate for large populations will poorly predict the epidemic dynamics of smaller populations since chance events may dramatically alter an epidemic trajectory when there are only a few cases to begin with [6]. As such, it is not clear that results describing the impacts of international travel restrictions will also apply within countries, to smaller regions, and to regions with low infection prevalence. Imported infections due to the arrival of infected travellers will have a disproportionately large effect when the number of local cases is few [5]. To appropriately characterize the impact of the travel restrictions on the COVID-19 outbreak in NL, we use a stochastic mathematical model appropriate for modelling infection dynamics in small populations [6], and where a similar modelling approach has been used in other jurisdictions [7,8]. Our analysis quantifies the impact of travel restrictions by considering a higher rate of imported infections when there are no travel restrictions, and we use the model to predict the number of cases that could have occurred in NL in the nine weeks subsequent to 4 May 2020.

# 2. Methods

## 2.1. Model overview

Our model is based on Plank *et al*. [7] who use a stochastic branching process to model COVID-19 dynamics in New Zealand. Our model describes the epidemiological dynamics of COVID-19 such that NL residents are susceptible to, infected with or recovered from COVID-19. Infected individuals are further divided into symptomatic and asymptomatic infections (infectious, no symptoms for the entire infectious period), and individuals with symptomatic infections may be in either the pre-clinical stage (infectious, prior to the onset of symptoms), or the clinical stage (infectious and symptomatic). The categorization of individuals into these infection classes is consistent with previous work [8,9].

Our model assumes that COVID-19 infections may spread when an infectious person contacts a susceptible person. Contact rates when physical distancing is undertaken in response to the pandemic are expressed in relative terms, as percentages of the contact rate relative to pre-pandemic levels. We assume that the pre-pandemic contact rate was equivalent to a basic reproduction number of $R_0 = 2.4$, where the definition of $R_0$ for our model is explained in table 1. Our model assumes that infected travellers that fail to self-isolate enter the population and may infect susceptible NL residents, and the rate of contact between residents and travellers is assumed to be the same as between residents. For individuals that are infectious (those with asymptomatic, pre-clinical and clinical infections), the probability of infection given a contact depends on the number of days since the date of infection [14], and infectivity further depends on whether the infection is pre-clinical, clinical or asymptomatic [9]. Individuals with clinical infections are relatively less infectious because these individuals are symptomatic and are more likely to self-isolate.

Similar to models developed by other researchers, our model is formulated as a continuous time branching process [7,8,16]. A branching process is a type of stochastic model where on any given simulation run, the predicted epidemic may be different since the epidemiological events, and the timing of these events, take values drawn from probability distributions. For example, our model assumes that the number of new infections generated by an infectious person follows a conditional Poisson distribution with a mean that depends on physical distancing, the number of susceptible individuals in the population, the type of infection the infected individual has (asymptomatic, pre-clinical or clinical), and the number of days since the date of infection (see electronic supplementary material, equation S1). Most other aspects of our model, for example, the timing of new infections, are similarly stochastic, each described by probability distributions that have appropriate characteristics, and are fully described in the electronic supplementary material. An overview of the model and all parameter values are given in figure 1 and table 1.

**Table 1.** Parameter values.

| quantity | description | source |
|---|---|---|
| *fixed* | | |
| $N_{pop} = 523\,000$ | NL population size | estimated as 519 716 in 2016 [10] |
| $\pi = 0.15$ | proportion of infections that are asymptomatic | estimated as 17% in Byambasuren et al. [11]. Known to take a wide range of values [12,13]. |
| $\eta_{JS} = 0.25$ | proportion reduction in infectivity for asymptomatic infections relative to clinical infections | [9] |
| $c_{iso} = 0.5$ | proportion reduction in infectivity for individuals with clinical infections due to self-isolation | [9] |
| $R_0 = 2.4$ | the pre-pandemic basic reproduction number. This is the number of secondary infections generated by an individual with a pre-clinical infection over their entire infectivity period, when all individuals in the population are susceptible. For our model, the definition of $R_0$ supposes that the level of infectivity corresponding to a pre-clinical infection is retained for the entire duration of the infectivity period (see the electronic supplementary material, equation S1). Note that a change in $R_0$ relative to its pre-pandemic level is equal to the same change in the contact rate if all other contributors to $R_0$ (i.e. the recovery rate and the probability of infection given a contact) are assumed unchanged | Assumed |
| $c_1 = 30\%$ | contact rate after 18 March 2020, expressed as a percentage of the pre-pandemic contact rate | Estimated from NL COVID-19 case data (figure 2) |
| $c_2$ in [40%, 70%] | contact rate after 4 May 2020, expressed as a percentage of the pre-pandemic contact rate | Range considered |
| *sampled* | | |
| $W(2.83, 5.67)$ | infectivity, which depends on the number of days since the date of infection (Weibull-distributed) | [14] |
| $s \sim \Gamma(6.1, 1.7)$ | the time from date of infection to self-isolation (gamma-distributed) | [7]. Note that $s \approx T_1 + T_2$ where $T_1$ and $T_2$ appear in [7]. |
| $z_1 \sim POIS(0.008)$ | the number of imported infected individuals per month that fail to self-isolate when travel restrictions are in place after 4 May 2020 (Poisson-distributed). The mean value is 0.24 infected travellers per month that fail to self-isolate. | Fit to NL COVID-19 case data when $c_2 \leq 60\%$ (figure 2) |
| $z_2 \sim POIS(0.1)$ | the number of imported infected individuals per month that fail to self-isolate when there are no travel restrictions after 4 May 2020 (Poisson-distributed). The mean value is three infected travellers per month that fail to self-isolate. | the mean importation rate is reduced by 92% when travel restrictions are in place since $z_1 = (1 - 0.92)z_2$. Therefore, the $z_2$ value is consistent with data reporting a 92.2% decrease in the number of passengers arriving at St. John's airport (NL) in June 2020 relative to June 2019 [15]. Equivalently, this assumption can be stated as that without travel restrictions the importation rate is 12.5 times greater (=0.1/0.008). |

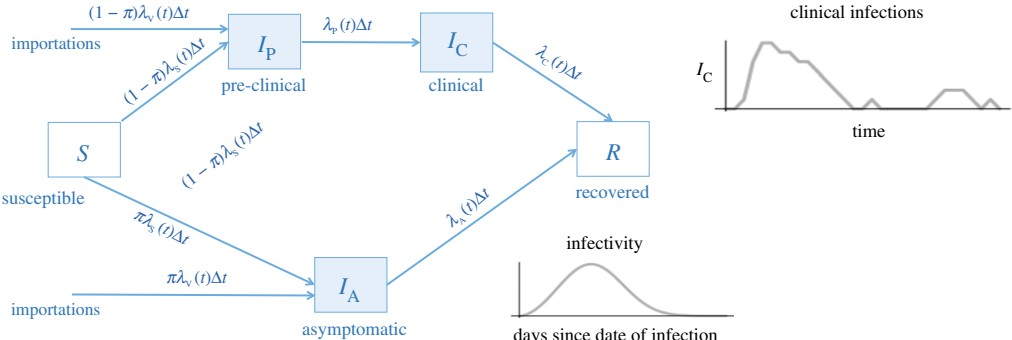

**Figure 1.** Model diagram. Uninfected individuals (white boxes) are either susceptible to infection, $S$, or recovered, $R$. Susceptible individuals become infected at mean rate, $\lambda_S(t)\Delta t$, where the event that an infection occurs is sampled from a distribution since the model is stochastic. Recovered individuals cannot be re-infected. Infected travellers that fail to self-isolate enter the population at a mean rate, $\lambda_V(t)\Delta t$. When a new infection occurs, a proportion, $\pi$, of these newly infected individuals are asymptomatic, where the number of individuals with asymptomatic infections at any time is $I_A$. Alternatively, a proportion, $1-\pi$, of infected individuals will eventually develop clinical symptoms, although these individuals are initially pre-clinical (without symptoms), and the number of individuals that are pre-clinical at any time is $I_P$. At a mean rate, $\lambda_P(t)\Delta t$, individuals with pre-clinical infections develop clinical infections (with symptoms). Individuals with asymptomatic, pre-clinical, and clinical infections are infectious (blue boxes), and infectivity depends on the type of infection, and the number of days since the date of infection. Finally, both individuals with asymptomatic and clinical infections recover at mean rates $\lambda_A(t)\Delta t$ and $\lambda_C(t)\Delta t$, respectively. See the electronic supplementary material for further details.

Our model does not consider age-structure or contact rates between individuals in the population that vary in space and time, due to, for example, attending school or work. This latter model limitation is discussed in the *Discussion* section. We intentionally limit the complexity of our model, since when additional parameters are added to a model the uncertainty in the predictions builds up, potentially to the point where the predictions may become useless [12]. The model is implemented in R and the code is publicly available at https://doi.org/10.6084/m9.figshare.12906710.v2.

### 2.1.1. Travel restriction scenarios

We assumed that the rate that infected individuals enter NL after 4 May and fail to self-isolate, is Poisson-distributed with a mean, $z_1 = 3$ (no travel restrictions) and $z_2 = 0.24$ per month (with travel restrictions). The assumed mean rate with travel restrictions yields model predictions compatible with the reported number of cases of COVID-19 in NL after 4 May 2020 (figure 2). These parameter values, $z_1$ and $z_2$, imply that with the travel restrictions the number of infected travellers arriving in NL and failing to self-isolate is reduced by 92%; or equivalently, without the travel restrictions the number of infected travellers arriving in NL and failing to self-isolate is 12.5 times greater. The mean rates that infected travellers enter NL and fail to self-isolate ($z_1$ and $z_2$) are compound parameters consisting of three components: (i) the rate that travellers enter NL, (ii) the proportion of travellers that are infected, and (iii) the proportion of infected travellers that fail to self-isolate. We do not resolve the individual contributions of these three components to $z_1$ and $z_2$; however, we note that only (i), the rate that travellers enter NL, probably changes when travel restrictions are in place. We assumed that infected travellers may be asymptomatic or pre-clinical, as symptomatic travellers are assumed to self-isolate. The proportion of infections that are asymptomatic is assumed to be the same for both travellers and NL residents. The mean rate that infected travellers enter NL is assumed to be constant over time and the origin cities of the travellers is not considered.

## 2.2. Epidemiological data and public health measures

From 14 March to 26 June 2020, the government of NL reported the number of active COVID-19 cases during media updates and on the [17] (for the relevant data, see also [18]). A copy of the data that was used for our analysis is archived with our code [19]. In addition to the travel restrictions enacted on 4 May 2020, legislation and public health recommendations that would have affected both the importation rate of COVID-19 to NL, and the spread of infections in the community are summarized

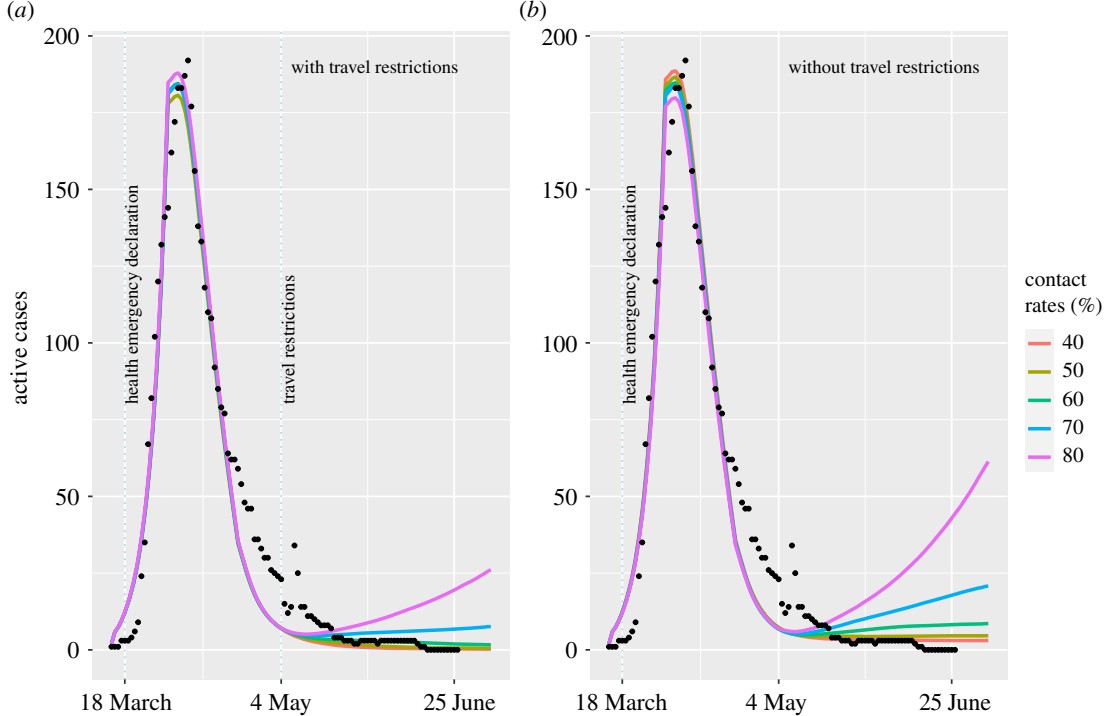

**Figure 2.** The predicted mean number of active COVID-19 cases (lines) agrees well with the reported numbers of active COVID-19 cases in NL from 16 March to 26 June 2020 (dots) prior to the implementation of the travel restrictions on 4 May 2020. After 4 May 2020, we consider an alternative past scenario where no travel restrictions were implemented (*b*). Both with (*a*) and without (*b*) the travel restrictions, we consider different levels of physical distancing, represented as percentages of the daily contact rate at the pre-pandemic level (coloured lines). Each coloured line is the mean number of active clinical cases each day calculated from 1000 runs of the stochastic model, which considers variability in the timing and changes in the number of individuals with different COVID-19 infection statuses.

in table 2. We assumed that the contact rate between NL residents changed after 18 March 2020, when a public health emergency was declared in NL.

### 2.2.1. Model calibration

We assumed that prior to 18 March 2020, the pre-pandemic basic reproduction number was $R_0 = 2.4$, where the assumed value of $R_0$ affects how quickly the epidemic would grow. All model parameters except the contact rate from 19 March to 4 May 2020, $c_1$, were estimated independently of the NL COVID-19 case data (table 1). The contact rate, $c_1$, is expressed as a percentage relative to the pre-pandemic contact rate (as implied by the pre-pandemic $R_0$ assuming all other contributors to $R_0$ are fixed). To fit $c_1$ given the data, we assumed that all clinical cases were reported, which is a reasonable assumption given the low number of cases reported in NL (for a model that considers unreported cases, see [23]). We estimated $c_1$ by observing that $c_1 = 30\%$ resulted in an agreement of the model with the epidemic data (further details of the model calibration are provided in the electronic supplementary material).

### 2.3. Output variables

To determine the impact of travel restrictions, we characterize clinical infections occurring in NL after 4 May 2020 as:

— Prior: the infected individual is part of an infection chain (i.e. a description of who infected whom) that originates from an NL resident infected prior to 4 May 2020.
— Travel: the infected individual was infected prior to travelling to NL.
— Local: the infected individual is an NL resident, who did not travel outside the province, and is part of an infection chain that originates from a traveller to NL.

**Table 2.** Public health measures implemented in Newfoundland and Labrador, 6 March–3 July 2020.

*measures affecting the importation rate*

| | |
|---|---|
| 20 March 2020 | all individuals arriving from outside NL must self-isolate for 14 days [20] |
| 4 May 2020 | — all individuals are prohibited from entering NL except:<br>　　a. residents of NL<br>　　b. asymptomatic workers and individuals subject to the Exemption Order.<br>　　c. individuals who have been permitted entry to NL, due to extenuating circumstances, approved in advance by the Chief Medical Officer of Health<br><br>— individuals arriving from outside NL must self-isolate for 14 days, be available for contact by public health, and complete a travel declaration form at the point of entry. (Special Measures Order—Travel, 15 May 2020). |
| 3 July 2020 | Atlantic bubble: interprovincial travel without the requirement to self-isolate permitted in Newfoundland and Labrador, New Brunswick, Prince Edward Island and Nova Scotia, for residents of Atlantic Canada [21]. |

*measures affecting community spread*

| | |
|---|---|
| 6 March 2020 | any resident with symptoms asked to stay at home and complete the self assessment tool [20] |
| 18 March 2020 | alert level 5*. State of emergency declared. Residents advised to practice physical distancing and only leave their homes for essential purposes. Only essential businesses open. Gatherings of more than 50 prohibited. Restaurants are takeout only. (Public Health Promotion and Protection Act; *inferred as alert levels not yet defined.) |
| 30 March 2020 | gatherings of more than five prohibited [20] |
| 11 May 2020 | alert level 4 [22]. Households are permitted to form 'double bubbles'. Gatherings of up to 10 people, reopening of parks and certain businesses. Childcare services operating at 50% [20] |
| 29 May 2020 | six more people can be added to 'double bubbles' [20] |
| 10 June 2020 | alert level 3. Gatherings of up to 20 people, responsible intra-provincial travel, and medium-intensity sports permitted. Childcare services operating at 70% [20] |
| 25 June 2020 | alert level 2. Occupancy and gatherings limited to 50 people, with physical distancing (including funerals, weddings, burials, indoor pools, gyms, movie theatres, bowling alleys, etc.). Wakes, karaoke and dance floors not allowed. Virtual delivery of health care encouraged. (Public Health Advisory 24 June 2020). |

The number of clinical cases that are 'travel-related' is calculated as the sum of infections characterized as 'travel' and 'local'. The predicted number of COVID-19 cases refers only to clinical infections, and does not include asymptomatic infections.

## 3. Results

The predicted number of active clinical COVID-19 cases in NL from 14 March to 4 May 2020 (figure 2, lines) broadly agrees with the data describing the number of active COVID-19 cases in NL over this same period (figure 2, black dots). From 4 May to 26 June 2020, when the travel restrictions were implemented in NL, the NL COVID-19 case data (figure 2a, black dots) agrees with the model predictions for physical distancing scenarios corresponding to contact rates ≤ 60% of the pre-pandemic level (figure 2a; coral, 40%; khaki, 50%; green, 60% lines).

We estimated that with the travel restrictions in place, from 4 May to 26 June 2020 the mean number of COVID-19 cases is reduced by 92% (table 3). For the different physical distancing scenarios considered, the mean number of cases over the nine weeks ranged from 14 to 48 clinical cases (without the travel restrictions), as compared with one to four clinical cases (with the travel restrictions; table 3 and figure 3a). These model predictions with the travel restrictions in place are

**Table 3.** Predicted total number of clinical COVID-19 cases in the nine weeks subsequent to 4 May 2020 with and without the implementation of travel restrictions. The prediction intervals represent the simulated 0.025 and 0.975 quantiles.

| percentage reduction in the contact rate relative to pre-pandemic levels | predicted clinical COVID-19 cases over nine weeks | | | |
| --- | --- | --- | --- | --- |
| | travel restrictions | no travel restrictions | magnitude greater without restrictions | percentage reduction with restrictions |
| **40%** | | | | |
| mean | 1.2 | 13.6 | 11.0 | 91.2% |
| median | 0 | 12 | | |
| 95% prediction intervals | [0,9] | [2,35] | | |
| **50%** | | | | |
| mean | 1.5 | 18.1 | 12.0 | 91.7% |
| median | 0 | 15 | | |
| 95% prediction intervals | [0,11] | [3,53] | | |
| **60%** | | | | |
| mean | 2.1 | 27.8 | 13.5 | 92.4% |
| median | 0 | 23 | | |
| 95% prediction intervals | [0,17] | [3,79] | | |
| **70%** | | | | |
| mean | 3.7 | 47.9 | 13.0 | 92.3% |
| median | 0 | 35 | | |
| 95% prediction intervals | [0,33] | [3,159] | | |
| | | | mean = 12.4 | mean = 91.9% |

consistent with the COVID-19 data for NL for the nine weeks following 4 May 2020 where during this time two new cases of COVID-19 were reported.

Without the travel restrictions, the number of clinical cases during the nine weeks can be very large (table 3 and figure 3a). Specifically, for a contact rate at 60% of its pre-pandemic level, the upper limit on the 95% prediction interval for the number of clinical cases over the nine weeks is 79 (without the travel restrictions) and 17 (with the travel restrictions; table 3 and figure 3a). The impact of the travel restrictions is even more substantial when only travel-related cases are considered (figure 3b) since almost all infections arising when the travel restrictions are implemented are attributed to infection chains that arise from an NL resident infected prior to 4 May. The mean number of cases of each infection type: 'prior', 'travel' and 'local' are shown in figure 4.

# 4. Discussion

Our model predictions broadly agree with the data describing the number of active COVID-19 cases in NL reported from 14 March to 4 May 2020, and from 4 May to 26 June 2020 if contact rates are 60% or less relative to pre-pandemic levels (figure 2). Our modelling shows that implementing the travel restrictions on 4 May 2020 reduced the number of COVID-19 cases by 92% over the subsequent nine weeks (table 3). Furthermore, without the travel restrictions, large outbreaks are much more likely (table 3, 95% prediction intervals; figure 3a). Travel restrictions alone may be insufficient to limit COVID-19 spread since the level of physical distancing undertaken by the local community, which affects the contact rates between residents, is also a strong determinant of the outbreak size (figures 2–4).

We found that the decrease in the mean number of clinical infections when the travel restrictions were enacted (a 92% reduction; table 3) was nearly exactly equal to the reduction in travel due to the travel restrictions (a 92% reduction; table 1). This equivalency was expected due to the hypothesized linear relationship between the importation rate and the mean outbreak size as noted in [24]. A consequence of this linear relationship is that any relative changes in the mean outbreak size are expected to be

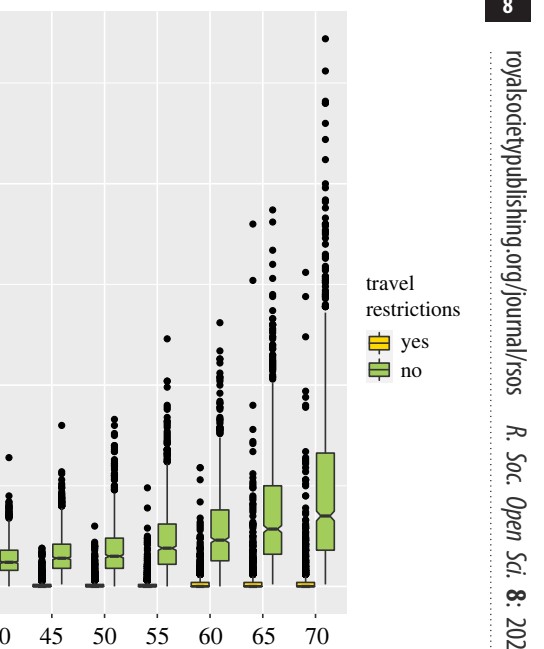

**Figure 3.** The total predicted number of COVID-19 cases in NL occurring over nine weeks beginning on 4 May 2020 when travel restrictions are implemented (yellow boxes) is much less than the total number of cases occurring over this same period if the travel restrictions were not implemented (green boxes). The total number of COVID-19 cases occurring during the nine weeks subsequent to 4 May 2020 is highly variable, and without the implementation of the travel restrictions there is a higher risk of a large outbreak (also see table 3, 95% prediction intervals). When the travel restrictions are implemented, almost all of the cases occurring during the nine weeks subsequent to 4 May 2020 are due to infected individuals present in the community prior to 4 May 2020. Travel-related cases are all cases remaining after the 'prior' cases are removed (*b*). The contact rate is expressed as a percentage of the pre-pandemic contact rate. For each simulation, chance events affect the number of individuals that change COVID-19 infection statuses and the timing of these changes. The horizontal lines are medians, the coloured boxes are 1.58 times the interquartile range divided by the square root of *n*, the whiskers are 95% prediction intervals, and the dots are outliers for the $n = 1000$ simulation outcomes.

equal to the relative changes in the importation rate (with travel restrictions relative to without restrictions and vice versa). The assumptions and characteristics of our model that give rise to this linear relationship are discussed in table 4 along with examples of conditions where these assumptions would be violated.

Related research, using North American airline passenger data from 1 January 2019 to 31 March 2020, in combination with epidemic modelling, found that depending on the type of travel restrictions, the effective reproduction number and the percentage of travellers quarantined, it would take between 37 and 128 days for 0.1% of the NL population to have been infected (table 2 in [25]). These predicted epidemic trajectories are consistent with our results. However, unlike [25], we have modelled importations and the NL epidemic dynamics as a stochastic process due to the low infection prevalence in NL at the time of our study.

## 4.1. Future directions

Our model does not consider spatial structure such that individuals contact each other in schools, workplaces or 'bubbles'. The absence of spatial structure in our model may over-estimate the probability of an epidemic establishing and the total number of cases until the outbreak subsides [26]. Related research, however, does consider spatially structured interactions in workplaces, businesses and schools, and concludes that without the travel restrictions implemented in NL on 4 May 2020 the number of COVID-19 cases would have been 10 times greater [27], which is in close agreement with our results that the number of cases would have been 12.5 times greater (table 3): a result that arises due to our parametrization of the importation rate without travel restrictions as 12.5 times greater than with travel restrictions (tables 1 and 4). Travel restrictions are one of several approaches available

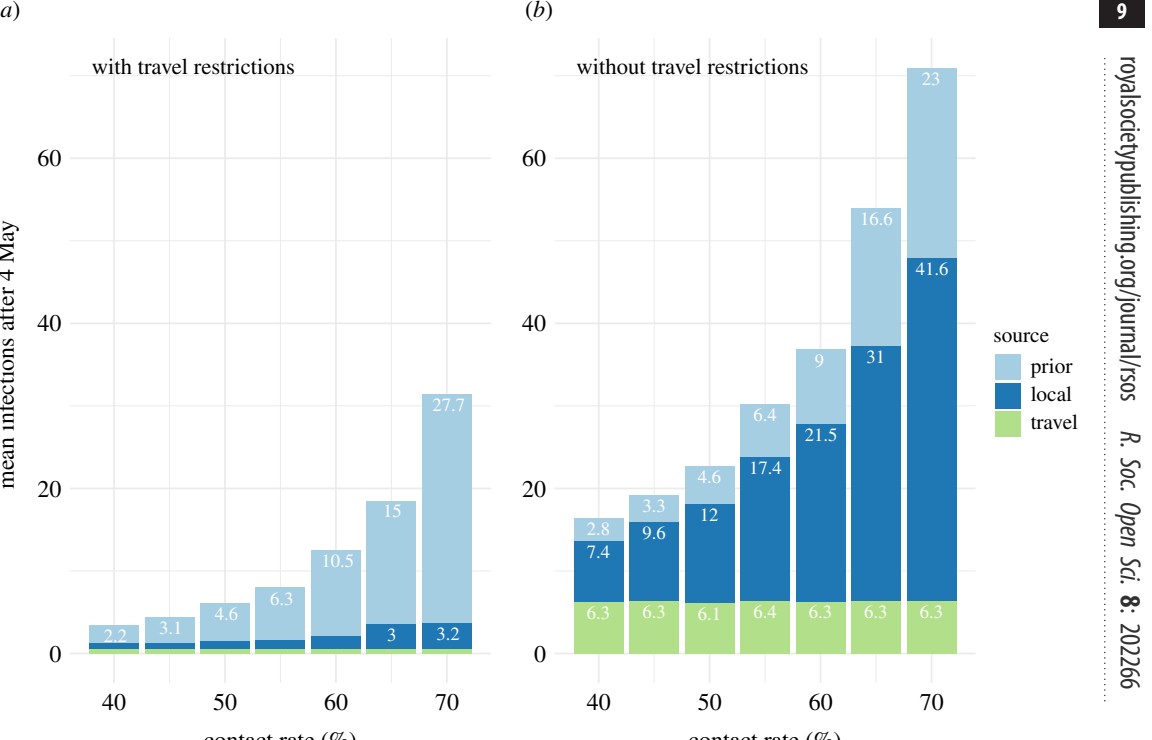

**Figure 4.** The breakdown into three different sources of COVID-19 cases occurring in NL over nine weeks. We compare simulation results with travel restrictions (*a*) and without travel restrictions (*b*). The source of infections is either: an individual infected prior to 4 May 2020 ('prior', light blue); an individual that was infected prior to entering NL ('travel', green); or an NL resident that did not travel, but is part of an infection chain where the initial infectee is a traveller that entered NL after 4 May 2020 ('local', dark blue). Our model assumptions are reflected by the difference in the number of COVID-19 cases occurring in travellers over the nine weeks (green bars): approximately 0.5 with travel restrictions (*a*), as compared with 6.3 without travel restrictions (*b*). These infected travellers seed infection chains in the NL community resulting in a larger number of NL residents infected when the travel restrictions are not implemented (dark blue bars). Both with and without the travel restrictions, the number of cases due to prior infection in the NL community is similar (light blue bars). The contact rate is expressed as a percentage of the pre-pandemic contact rate.

to health authorities for COVID-19 management. Future research should consider the role of travel restrictions, testing, contact tracing and physical distancing, as elements of comprehensive approach to the best management of COVID-19.

## 4.2. Limitations

We were not able to estimate the rate that infected travellers enter NL; however, other research modelling infection dynamics in the origin cities of air travellers to NL found that without travel restrictions a new COVID-19 case would enter NL every other day [25]. Similarly, we were not able to estimate the percentage of travellers to NL that comply with self-isolation directives. Smith *et al.* [28] found that 75% of survey participants reporting COVID-19 symptoms (high temperature and/or cough) also report having left their house in the last 24 h, violating the lockdown measures in place in the UK at the time, and so non-compliance rates may be quite high. Our analysis does not consider hospitalizations or deaths; however, we note that as of 4 May 2020, NL had experienced 259 clinical cases and three deaths. With the contact rate at 80% of its pre-pandemic level and no travel restrictions, we estimate that it would take, on average, 10.2 weeks for a further 259 clinical cases to occur, and although there is evidence that case fatality rates have changed over time [29], it is reasonable to expect a further three deaths under these conditions. By contrast, with the travel restrictions in place, it would take more than six months (28.1 weeks) for this same number of cases and deaths to accumulate. Thus, with the first COVID-19 vaccines available to the public a year after the beginning of the pandemic, the value of enacting travel restrictions to delay the local outbreak by six months is potentially substantial.

**Table 4.** A list of the assumptions and characteristics of our model that give rise to the linear relationship between the importation rate and the mean outbreak size. The linear relationship is that $I_{tot} = \lambda_v I_1$, where $I_{tot}$ is the mean total number of cases, $\lambda_v$ is the importation rate, and $I_1$ is the mean number of cases that arise from one importation.

| model assumption | example where the model assumption is violated | effect of violating the assumption on outbreak size |
|---|---|---|
| Mixing between individuals in the population is homogeneous. | A group of travellers, all of whom are infected, fail to self-isolate, but also travel everywhere together and contact all of the same people. | No matter what the size of the group, the resulting outbreak will be of similar size since the contacts of group members are redundant. Here, the mean outbreak size is not linearly related to the importation rate because a larger group would correspond to a larger number of importations, yet the resulting outbreak would not be much larger. |
| Homogeneous mixing means than an infected person is equally likely to contact every susceptible person in the population. | Mixing is non-homogeneous because group members are constrained to have contacts only among the same individuals as the other group members, and not all individuals in the population. | |
| The number of susceptible people is relatively unchanged during the timeframe of interest. | The susceptible population is small, or infection control measures are few. | Infected individuals that arrive later will generate smaller infection chains due to fewer susceptible people to infect. Therefore, the total outbreak size cannot be calculated by summing the size of the outbreaks per importation, since the timing of the importation affects the outbreak size due to that importation. |
| The number of people an infected person contacts is unchanged during the timeframe of interest. | Waning compliance with public health measures; school re-openings. | As above, outbreak sizes per importation cannot cannot be added to determine the total outbreak size because the timing of the importations affects the value of the outbreak size per importation. |
| Infectivity does not change over time. | Seasonality | See above. |
| **model characteristic** | **a different characteristic** | **effect of considering the different characteristic** |
| Few 'prior' cases: cases that are not attributable to importations (see *Methods—Output variables*). | High infection prevalence in the absence of importations. | The relationship between travel-related cases and the importation rate will be linear, but total infections are the sum of prior cases and travel-related cases, such that the linear relationship will not hold. |
| The quantity of interest is the mean outbreak size. | The quantity of interest is the median or a different quantile. | The linear relationship with the importation rate applies only to the mean outbreak size. As can be observed in table 2, the linear relationship does not apply to the median or 95% prediction intervals. |

## 4.3. Conclusion

At the time of the implementation of the travel restrictions, there were few COVID-19 infections in NL. Without the travel restrictions, most of the subsequent COVID-19 infections would have been initiated by infected travellers who failed to comply with self-isolation requirements and only the actions of NL residents (i.e. physical distancing), and local health authorities (i.e. testing and contact tracing) would be sufficient to slow the exponential growth of these infection chains in the local community.

Data accessibility. All data and code are archived in figshare: https://doi.org/10.6084/m9.figshare.12906710.v2.
The data are provided in electronic supplementary material [30].

Authors' contributions. A.H. conceptualized the analysis, interpreted the results and wrote the manuscript. J.C.L.-O. conceptualized the analysis, wrote the code, made the figures, interpreted the results and revised the analysis. P.R. interpreted the results and revised the manuscript.

Competing interests. All authors are members of the Predictive Analytics team assembled by the Newfoundland and Labrador Center for Health Information to provide mathematical modelling support to the province of NL. This manuscript arises from a technical report that was prepared at the request of the Department of Justice and Public Safety, NL. This study was not influenced by any representatives of the province of NL.

Funding. A.H. was supported by a Discovery Grant from the National Sciences and Engineering Research Council of Canada.

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
