## [Peer Review File · Royal Society Open Science]

Review History

RSOS-202266.R0 (Original submission)

Review form: Reviewer 1

Is the manuscript scientifically sound in its present form?

Yes

Are the interpretations and conclusions justified by the results?

Yes

Is the language acceptable?

Yes

Do you have any ethical concerns with this paper?

No

Have you any concerns about statistical analyses in this paper?

No

Recommendation?

Accept with minor revision (please list in comments)

Comments to the Author(s)

See attached file (Appendix A).

Review form: Reviewer 2

Is the manuscript scientifically sound in its present form?

No

Are the interpretations and conclusions justified by the results?

Yes

Is the language acceptable?

Yes

Do you have any ethical concerns with this paper?

No

Have you any concerns about statistical analyses in this paper?

No

Recommendation?

Major revision is needed (please make suggestions in comments)

Comments to the Author(s)

Review comments for the manuscript, "Modelling the impact of travel restrictions on COVID-19 cases in Newfoundland and Labrador" by Hurford et al.

A stochastic disease model is fitted to COVID-19 case data from Newfoundland and Labrador. The calibrated model is used to predict outbreaks when travel restrictions are actively in place and when there are no active travel restrictions. The main result is that travel restrictions have a positive impact in reducing the number of clinical disease cases. In my opinion, the main result is obvious. To be precise, I don't think that this study is a significant contribution to an already saturated area. Some specific comments are provided below.

Limiting the data used in the study to May 4, 2020 (i.e., barely 49 days), despite the fact that there is currently more data restricts the strength of predictions and study.

It is not clear why the pre-clinical (pre-symptomatic) class feeds only to the symptomatic class. The asymptomatic are those who do not exhibit disease symptoms after the incubation period. But some of those who become asymptomatic after the incubation period can also start transmitting after the latent period and before the end of the incubation period.

Why are the authors tracking only the pre-symptomatic and asymptomatic individuals who enter the community? How about the other classes and pre-symptomatic and asymptomatic individuals who leave the community?

How the model was fitted to data is not explained.

Decision letter (RSOS-202266.R0)

Dear Dr Hurford

On behalf of the Editors, we are pleased to inform you that your Manuscript RSOS-202266 "Modelling the impact of travel restrictions on COVID-19 cases in Newfoundland and Labrador" has been accepted for publication in Royal Society Open Science subject to minor revision in accordance with the referees' reports. Please find the referees' comments along with any feedback from the Editors below my signature.

Please submit your revised manuscript and required files (see below) no later than 7 days from today's (ie 22-Apr-2021) date. Note: the ScholarOne system will 'lock' if submission of the revision is attempted 7 or more days after the deadline. If you do not think you will be able to meet this deadline please contact the editorial office immediately.

Best regards,

on behalf of Dr Pierre Magal (Associate Editor) and Pete Smith (Subject Editor)
openscience@royalsociety.org

Associate Editor Comments to Author (Dr Pierre Magal):

I think the paper should be accepted with minor revisions.

Please try to follow as much as possible the suggestions of the referee.

In the case of the second referee please consider the following point: "Why are the authors tracking only the pre-symptomatic and asymptomatic individuals who enter the community? How about the other classes and pre-symptomatic and asymptomatic individuals who leave the community?"

How the model was fitted to data is not explained."

Reviewer comments to Author:

Reviewer: 1
Comments to the Author(s)

See attached file

Reviewer: 2
Comments to the Author(s)

Review comments for the manuscript, "Modelling the impact of travel restrictions on COVID-19 cases in Newfoundland and Labrador" by Hurford et al.

A stochastic disease model is fitted to COVID-19 case data from Newfoundland and Labrador. The calibrated model is used to predict outbreaks when travel restrictions are actively in place and when there are no active travel restrictions. The main result is that travel restrictions have a positive impact in reducing the number of clinical disease cases. In my opinion, the main result is obvious. To be precise, I don't think that this study is a significant contribution to an already saturated area. Some specific comments are provided below.

Limiting the data used in the study to May 4, 2020 (i.e., barely 49 days), despite the fact that there is currently more data restricts the strength of predictions and study.

It is not clear why the pre-clinical (pre-symptomatic) class feeds only to the symptomatic class. The asymptomatic are those who do not exhibit disease symptoms after the incubation period. But some of those who become asymptomatic after the incubation period can also start transmitting after the latent period and before the end of the incubation period.

Why are the authors tracking only the pre-symptomatic and asymptomatic individuals who enter the community? How about the other classes and pre-symptomatic and asymptomatic individuals who leave the community?

How the model was fitted to data is not explained.

===PREPARING YOUR MANUSCRIPT===

Your revised paper should include the changes requested by the referees and Editors of your manuscript. You should provide two versions of this manuscript and both versions must be provided in an editable format:
one version identifying all the changes that have been made (for instance, in coloured highlight, in bold text, or tracked changes);
a 'clean' version of the new manuscript that incorporates the changes made, but does not highlight them. This version will be used for typesetting.
Please ensure that any equations included in the paper are editable text and not embedded images.

===PREPARING YOUR REVISION IN SCHOLARONE===

Author's Response to Decision Letter for (RSOS-202266.R0)

See Appendix B.

Decision letter (RSOS-202266.R1)

Dear Dr Hurford:

I write you in regards to manuscript # RSOS-202266.R1 entitled "Modelling the impact of travel restrictions on COVID-19 cases in Newfoundland and Labrador" which you submitted to Royal Society Open Science.

Regrettably, in view of the criticisms of the reviewer(s) found at the bottom of this letter, your manuscript has been denied publication in Royal Society Open Science.

Thank you for considering Royal Society Open Science for the publication of your research. I hope the outcome of this specific submission will not discourage you from the submission of future manuscripts.

on behalf of Dr Pierre Magal (Associate Editor) and Pete Smith (Subject Editor)
openscience@royalsociety.org

Associate Editor Comments to Author (Dr Pierre Magal):
Associate Editor
Comments to the Author:

I don't see a single equation written in the paper, while everything done in this paper is based on that. Therefore, their work is not reproducible which is the first requirement of scientific work.

Next the idea of asymptomatic patient was introduced first for COVID-19 outbreak in

Z. Liu, P. Magal, O. Seydi, and G. Webb (2020), Understanding unreported cases in the 2019-nCoV epidemic outbreak in Wuhan, China, and the importance of major public health interventions, *Biology* 9(3), 50.

This article (and none of the following papers is not quote), therefore I don't think this paper is suitable for publication in Royal Society Open Sciences.

Reviewer comments to Author:

Author's Response to Decision Letter for (RSOS-202266.R1)

See Appendix C.

Decision letter (RSOS-202266.R2)

Dear Dr Hurford,

I am pleased to inform you that your manuscript entitled "Modelling the impact of travel restrictions on COVID-19 cases in Newfoundland and Labrador" is now accepted for publication in Royal Society Open Science.

COVID-19 rapid publication process:

We are taking steps to expedite the publication of research relevant to the pandemic. If you wish, you can opt to have your paper published as soon as it is ready, rather than waiting for it to be published the scheduled Wednesday.

This means your paper will not be included in the weekly media round-up which the Society sends to journalists ahead of publication. However, it will still appear in the COVID-19 Publishing Collection which journalists will be directed to each week (<https://royalsocietypublishing.org/topic/special-collections/novel-coronavirus-outbreak>).

If you wish to have your paper considered for immediate publication, or to discuss further, please notify openscience_proofs@royalsociety.org and press@royalsociety.org when you respond to this email.

on behalf of Dr Pierre Magal (Associate Editor) and Pete Smith (Subject Editor)
openscience@royalsociety.org

Appendix A

This is an interesting study that effectively quantifies the importance of restricting arrivals to limit the spread of Sars-Cov-2 caused by further reintroductions. Whilst this may seem obvious it is definitely worth saying as countries like the UK only now implement moderate restrictions. The paper is well-written and easy-to-follow. The model is simple but appropriate given the lack of more specific data, in particular contact rates.

The work doesn't mention the original data beyond the black dots in Figure 2. It would be useful to see more specific comparisons with the data beyond this. For example, how many cases occurred before and after the travel ban, how many does the model predict. This could also help in estimating the actual reduction in contact rates. With a short summary of the restrictions that were in place over this period this would make the paper far more useful for international efforts to predict the effect of non-pharmaceutical interventions to control the spread of Sars-Cov-2.

Specific comments:

You assume travellers have the same contact rates as residents. This may depend on the type of traveller, i.e. returning resident, commuter, tourist etc.

Spread of Sars-Cov-2 is often driven by superspreading events but you use a Poisson distribution for secondary infections rather than a negative binomial (see Lloyd-Smith, Nature). Would this change your results?

From Fig 1 the model seems to assume that infected arrivals are pre-clinical rather than clinical. Is this justified?

Fig 1: top right "Model predictions". It's not clear what this picture is! Please label sub figures and give more details in the caption.

Appendix B

Associate Editor Comments to Author (Dr Pierre Magal):

I think the paper should be accepted with minor revisions.

Thank you for efforts in evaluating our manuscript. Our responses are in bold. To summarize the changes made to the manuscript:

- We have made text edits to clarify ambiguities as raised by the reviewers.
- We had added details of the model fitting in response to the AE and Reviewer 2 (Figure A.3 added).
- We have added details pertaining to the data in response to Reviewer 1.
- We have added Table 2, which summarizes non-pharmaceutical interventions implemented in NL during the study period at the suggestion of Reviewer 1.
- We repeated our simulations using a negative binomial distribution of secondary infections in response to Reviewer 1 (Figures A.1 and A.2 added).

Please try to follow as much as possible the suggestions of the referee.

In the case of the second referee please consider the following point: "Why are the authors tracking only the pre-symptomatic and asymptomatic individuals who enter the community?"

AE Response 1. We assumed that symptomatic individuals would isolate and not enter the community. This is consistent with public health guidelines for Newfoundland and Labrador (NL) during the study period. This is also consistent with our assumption that symptomatic individuals in the community have a low contact rate because they are likely to self isolate. To clarify this point we made the following change:

rate that travellers enter NL, likely changes when travel restrictions are in place. We assumed that infected travellers may be asymptomatic or pre-clinical, as symptomatic travellers are assumed to self-isolate. The proportion of infections that are asymptomatic is assumed to be the same for both

How about the other classes and pre-symptomatic and asymptomatic individuals who leave the community?

Regarding 'other classes', latently infected individuals are captured within the pre-symptomatic and asymptomatic classes because infectivity changes as a function of time since infection following the Weibull distribution. When few days have passed since infection, then infectivity is low, similar to a latent class. Rather than having discrete classes, we have infectivity as a continuous function of time since infection, and as referenced in the manuscript this formulation was based on the models derived by several leading research groups, and captures the appropriate epidemiology. We do not consider spatial structure (this sensitivity is discussed in the Discussion) or age structure due to the lack of data on age specific contact rates for NL.

Regarding individuals who leave the community, with travel severely restricted at the time, and even our worst cases scenario having only around 60 active cases (Figure 2b) which is ~0.01% of the NL population, it seems unlikely that a pre-symptomatic or asymptomatic individuals would leave. Further, if a pre-symptomatic or asymptomatic individual did leave, it seems unlikely that this would substantially alter the exponential growth trajectory. Therefore, we did not consider pre-symptomatic and asymptomatic individuals leaving the community to be a salient model feature.

How the model was fitted to data is not explained."

AE Response 2. The subheading "Model calibration" has been added to the manuscript, and additional details of the model calibration are provided in the Appendix (see also Figure A.3).

Reviewer comments to Author:

Reviewer: 1

Comments to the Author(s)

[Copy and paste from pdf]

This is an interesting study that effectively quantifies the importance of restricting arrivals to limit the spread of Sars-Cov-2 caused by further reintroductions. Whilst this may seem obvious it is definitely worth saying as countries like the UK only now implement moderate restrictions. The paper is well-written and easy-to-follow. The model is simple but appropriate given the lack of more specific data, in particular contact rates.

Thank you. The reviewer accurately describes our aims in writing this manuscript.

The work doesn't mention the original data beyond the black dots in Figure 2. It would be useful to see more specific comparisons with the data beyond this. For example, how many cases occurred before and after the travel ban, how many does the model predict. This could also help in estimating the actual reduction in contact rates.

We included more details describing the data and the agreement of the model with the data before and after the travel restrictions. Specific edits in response to this suggestion are:

Epidemiological data and public health measures

From March 14th to June 26th, 2020, the government of NL reported the number of active COVID-19 cases during media updates and on the Newfoundland and Labrador Pandemic Update Data Hub (for the relevant data, see also Berry et al. 2020). A copy of the data that was used for our analysis is archived with our code (Hurford, Rahman, and Loredo-Osti 2020). In addition to the travel restrictions enacted on May 4th, legislation and public health recommendations that would have affected both the importation rate of COVID-19 to NL, and the spread of infections in the community are summarized in Table 2. We assumed that the contact rate between NL residents changed after March 18, 2020, when a public health emergency was declared in NL.

The predicted number of active clinical COVID-19 cases in NL from March 14th to May 4th (Figure 2, lines) broadly agrees with the data describing the number of active COVID-19 cases in NL over this same period (Figure 2, black dots). From May 4th to June 26, 2020, when the travel restrictions were implemented in NL, the NL COVID-19 case data (Figure 2a, black dots) agrees with the model predictions for physical distancing scenarios corresponding to contact rates \leq 60% of the pre-pandemic level (Figure 2a; coral – 40%, khaki - 50%, and green – 60% lines).

Discussion

Our model predictions broadly agree with the data describing the number of active COVID-19 cases in NL reported from March 14th to May 4th, and from May 4th to June 26th if contract rates are 60% or less relative to pre-pandemic levels (Figure 2). Our modelling shows that implementing the travel

With a short summary of the restrictions that were in place over this period this would make the paper far

more useful for international efforts to predict the effect of non-pharmaceutical interventions to control the spread of Sars-Cov-2.

This is an excellent suggestion. We have added Table 2, a short summary of the non-pharmaceutical interventions implemented in NL during the period of our study.

Specific comments:

You assume travellers have the same contact rates as residents. This may depend on the type of traveller, i.e. returning resident, commuter, tourist etc.

Agree, but we lack the data to be able to parameterize this aspect of the model well.

Spread of Sars-Cov-2 is often driven by superspreading events but you use a Poisson distribution for secondary infections rather than a negative binomial (see Lloyd-Smith, Nature). Would this change your results?

Assuming the negative binomial did not change our results. We have added a section addressing this point in the Appendix.

From Fig 1 the model seems to assume that infected arrivals are pre-clinical rather than clinical. Is this justified?

We have clarified in the text that we assumed clinical (symptomatic) arrivals undergo self-isolation.

Fig 1: top right "Model predictions". It's not clear what this picture is! Please label sub figures and give more details in the caption.

"Model predictions" has been renamed "Clinical infections".

Reviewer: 2

Comments to the Author(s)

Review comments for the manuscript, "Modelling the impact of travel restrictions on COVID-19 cases in Newfoundland and Labrador" by Hurford et al.

A stochastic disease model is fitted to COVID-19 case data from Newfoundland and Labrador. The calibrated model is used to predict outbreaks when travel restrictions are actively in place and when there are no active travel restrictions. The main result is that travel restrictions have a positive impact in reducing the number of clinical disease cases. In my opinion, the main result is obvious. To be precise, I don't think that this study is a significant contribution to an already saturated area. Some specific comments are provided below.

As noted by Reviewer 1, the value of travel restrictions may seem obvious, but in countries such as the UK, only now have moderate restrictions been implemented. In Newfoundland and Labrador, the focus of our study, the travel restrictions implemented on May 4th 2020, were challenged in the Supreme Court of Newfoundland and Labrador (decision rendered September 17, 2020). Therefore, we feel that this work is very necessary, because a very strong scientific argument is required for policy makers to justify the implementation of travel restrictions.

Limiting the data used in the study to May 4, 2020 (i.e., barely 49 days), despite the fact that there is currently more data restricts the strength of predictions and study.

The data are from March 16th to June 26th (not May 4th). Text edits have been made to clarify this point. This study was first undertaken to meet a June 30th deadline. While further data are available now, the period May 4th to June 26th is representative of the subsequent epidemic dynamics in NL (a few travel-related cases and no community outbreaks) even through to the end of 2020.

It is not clear why the pre-clinical (pre-symptomatic) class feeds only to the symptomatic class. The asymptomatic are those who do not exhibit disease symptoms after the incubation period. But some of those who become asymptomatic after the incubation period can also start transmitting after the latent period and before the end of the incubation period.

Asymptomatic individuals are not defined as ‘those who do not exhibit disease symptoms after the incubation period’ as the reviewer assumes. Please see the definition of asymptomatic individuals as stated in the manuscript:

such that NL residents are either susceptible to, infected with, or recovered from COVID-19. Infected individuals are further divided into symptomatic and asymptomatic infections (infectious, no symptoms for the entire infectious period), and individuals with symptomatic infections may be in

and in the Figure 1 caption:

develop clinical infections (with symptoms). Individuals with asymptomatic, pre-clinical, and clinical infections are infectious (blue boxes), and infectivity depends on the type of infection, and the

Why are the authors tracking only the pre-symptomatic and asymptomatic individuals who enter the community? How about the other classes and pre-symptomatic and asymptomatic individuals who leave the community?

Please see AE response 1.

How the model was fitted to data is not explained.

Please see AE response 2.

Appendix C

Subject Editor Comments to Author (Prof Pete Smith)

A different AE handled the revised submission. Whilst the AE recommends rejection, I think that given your previous decision and your subsequent revisions, we should accept the paper. The AE suggests additional literature that should be included. Please consider adding this (and any subsequent papers), after which your paper will be ready for publication.

AH: Our sincere thanks to the subject editor for evaluating the reviews for this manuscript and making a decision based on our genuine effort to address the comments from the previous round of review.

Associate Editor Comments to Author (Dr Pierre Magal):

Comments to the Author:

I don't see a single equation written in the paper, while everything done in this paper is based on that. Therefore, their work is not reproducible which is the first requirement of scientific work.

AH: The equations are in the Electronic Supplementary Material as indicated in Methods overview section paragraph 3. In addition, the code is archived at <https://doi.org/10.6084/m9.figshare.12906710.v2>. as indicated in the last sentence of the same section.

Next the idea of asymptomatic patient was introduced first for COVID-19 outbreak in

Z. Liu, P. Magal, O. Seydi, and G. Webb (2020), Understanding unreported cases in the 2019-nCoV epidemic outbreak in Wuhan, China, and the importance of major public health interventions, *Biology* 9(3), 50.

AH: This paper has now been cited in the manuscript (see the Tracked changes versions – *model calibration* section).

This article (and none of the following papers) is not quoted)

AH: I am unclear on what the AE is recommending. Regarding the subject editor's invitation to add more literature, we have added Grépin et al. 2021.